# Inference with correlated priors using sisters cells

**Sina Tootoonian and Andreas T. Schaefer**
Sensory Circuits and Neurotechnology Laboratory
The Francis Crick Institute
London, UK
[sina.tootoonian|andreas.schaefer]@crick.ac.uk

## Abstract

A common view of sensory processing is as probabilistic inference of latent causes from receptor activations. Standard approaches often assume these causes are *a priori* independent, yet real-world generative factors are typically correlated. Representing such structured priors in neural systems poses architectural challenges, particularly when direct interactions between units representing latent causes are biologically implausible or computationally expensive. Inspired by the architecture of the olfactory bulb, we propose a novel circuit motif that enables inference with correlated priors without requiring direct interactions among latent cause units. The key insight lies in using *sister cells*: neurons receiving shared receptor input but connected differently to local interneurons. The required interactions among latent units are implemented indirectly through their connections to the sister cells, such that correlated connectivity implies anti-correlation in the prior and vice versa. We use geometric arguments to construct connectivity that implements a given prior and to bound the number of causes for which such priors can be constructed. Using simulations, we demonstrate the efficacy of such priors for inference in noisy environments and compare the inference dynamics to those experimentally observed. Finally, we show how, under certain assumptions on latent representations, the prior used can be inferred from sister cell activations. While biologically grounded in the olfactory system, our mechanism generalises to other natural and artificial sensory systems and may inform the design of architectures for efficient inference under correlated latent structure.

## 1   Introduction

A common view of sensory processing is as probabilistic inference of latent causes,

$$\mathbf{x} \triangleq [x_1, \ldots, x_N],$$

from receptor inputs [1]. Causes are often assumed to be *a priori* independent [2], so that

$$p(\mathbf{x}) = p(x_1, x_2, \ldots, x_N) \propto \prod_{i=1}^{N} e^{-\phi_i(x_i)}.$$

This assumption is not only mathematically convenient, but is also appropriate in some settings. For example, in the celebrated 'cocktail party problem,' signals from multiple microphones must be demixed into as many simultaneous conversations, and it is reasonable to assume that the audio waveforms from different conversations will be independent.

In other situations, however, the independence assumption may not be appropriate. For example, some notable models [3, 4] of the early visual system describe it as explaining retinal input in terms of simple features like oriented Gabors. Although they arrive at such features by searching for

39th Conference on Neural Information Processing Systems (NeurIPS 2025).

independent causes that explain the visual input, natural scenes are likely to impose correlations on the presence of such features due to large-scale visual structures.

Including correlations is in principle easy to do by incorporating corresponding terms in the prior. For example, pairwise correlations, the focus of our present work, can be incorporated by augmenting the prior with quadratic terms,

$$p(x_1, x_2, \ldots, x_n) \propto \exp\left(-\sum_i \phi_i(x_i) - \frac{1}{2}\sum_{ij} C_{ij}x_ix_j\right).$$

Non-zero off-diagonal elements of $C_{ij}$ capture correlations among the corresponding features. A prior of this form, which sets $\phi_i(x_i)$ proportional to $x_i$, is the Gaussian Markov random field [5].

Although analytically simple to incorporate, neural implementations of inference circuits that use such priors can pose architectural challenges. To illustrate this, we will use a simple model of the mammalian olfactory bulb, from which we take inspiration. In this model, receptors $y_i$ are linearly excited by latent features $x_j$ (e.g. molecular species comprising an odour), and corrupted by Gaussian noise, so that $p(y_i|\mathbf{x}) = \mathcal{N}\left(\sum_j A_{ij}x_j, \sigma^2\right)$. Combining this with the prior above yields the posterior distribution, whose logarithm

$$\log p(x_1, \ldots, x_N|y_1, \ldots, y_M) = \underbrace{-\sum_j \phi_j(x_j) - \sum_{jk} C_{jk}x_jx_k}_{\text{log prior}} \underbrace{-\sum_i \frac{1}{2\sigma^2}(y_i - \sum_j A_{ij}x_j)^2}_{\text{log likelihood}} \quad (1)$$

can be maximized over features to yield the *maximum a posteriori* (MAP) estimate of the combination of features (e.g. odour) most likely to have produced the observed receptor activations. A circuit that performs this maximization (see e.g. [6]) contains 'mitral cells' $\lambda_i$, one per input channel, that compare actual receptor inputs $y_i$ with the system's current estimate, $\sum_j A_{ij}x_j$,

$$\tau_\lambda \dot{\lambda}_i = -\sigma^2 \lambda_i + y_i - \sum_j A_{ij}x_j, \quad (2)$$

interacting with 'granule cells' representing the latent causes $x_j$. Granule cells are driven by the mitral cells, and are subject to a prior that includes both the individual terms $\phi'_j(x_j) \triangleq d\phi_j(x_j)/dx_j$ and the pairwise terms $C_{jk}$,

$$\tau_x \dot{x}_j = -\phi'_j(x_j) - \sum_k C_{jk}x_k + \sum_i A_{ij}\lambda_i. \quad (3)$$

It is easy to see that the fixed point of the coupled dynamics in Eqn. (2) and Eqn. (3) maximizes the log posterior in Eqn. (1). When the dynamics converge, the activity of the granule cells represents the inferred concentrations of the molecules represented by each granule cell.

The problem with this formulation is that each latent cause must interact with many others, as determined by the elements $C_{jk}$. Such connectivity may be difficult to implement when the number of causes $N$ is very large. For example, in our model system of the olfactory bulb we assume that latent causes are represented by the millions of granule cells, and no direct connections have been observed between them [7]. These points suggest looking elsewhere to implement the pairwise interactions in the prior.

## 2 Encoding correlated priors using sister cells

To encode correlated priors without direct interaction between granule cells representing latent causes, we note that in the olfactory bulb, each mitral cell has many sisters — other mitral cells that receive the same receptor input but connect differently to granule cells [7–9]. We now show how correlated priors can be encoded indirectly by the way granule cells connect to the sister cells.

In the dynamics of Eqn. (2) and Eqn. (3), a mitral cell $\lambda_i$ is indexed $i$ by the receptor input $y_i$ it receives. To extend these dynamics to sister cells, we simply add an index $s$ to identify individual sisters, and endow each sister $\lambda_{is}$ with its own connectivity $A^s_{ij}$ to the granule cells:

$$\tau_\lambda \dot{\lambda}_{is} = -\sigma^2 \lambda_{is} + y_i - \sum_j A^s_{ij}x_j. \quad (4)$$

The granule cell dynamics in Eqn. (3) are correspondingly changed to drop the latent interaction term and instead distinguish sister cells

$$\tau_x \dot{x}_j = -\phi'_j(x_j) + \sum_i \sum_s A^s_{ij} \lambda_{is}. \tag{5}$$

Although we derived these dynamics heuristically by analogy to the case of a single mitral cell per input channel, we show in Sec. S1 that their fixed points minimize the loss

$$\mathcal{L}(\mathbf{x}) = \sum_j \phi_j(x_j) + \sum_i \sum_s \frac{1}{2\sigma^2} (y_i - \sum_j A^s_{ij} x_j)^2. \tag{6}$$

We can give a probabilistic interpretation to this loss by first defining a few connectivity statistics. Letting $S_i$ indicate the number of sister cells receiving receptor input $y_i$, we define

$$\overline{A}_{ij} \triangleq \frac{1}{S_i} \sum_s A^s_{ij}, \quad C^i_{jk} \triangleq \frac{1}{S_i} \sum_s A^s_{ij} A^s_{ik} - \overline{A}_{ij} \overline{A}_{ik}, \quad C_{jk} \triangleq \sum_i S_i C^i_{jk} \tag{7}$$

as the average synaptic weight from sisters to granule cell $x_j$, the covariance of the weights which connect two granule cells $x_j$ and $x_k$ to a set of sisters, and the weighted sum of these covariances across input channels, respectively. Expressing the loss in Eqn. (6) using these terms and completing the square (see Sec. S1) we arrive at

$$\mathcal{L}(\mathbf{x}) = \underbrace{\sum_j \phi_j(x_j) + \frac{1}{2\sigma^2} \sum_j \sum_k C_{jk} x_j x_k}_{-\log \text{ prior}} + \underbrace{\frac{1}{2\sigma^2} \sum_i S_i (y_i - \sum_j \overline{A}_{ij} x_j)^2}_{-\log \text{ likelihood}}. \tag{8}$$

This has the same form as the loss in Eqn. (1), demonstrating that sister cell dynamics perform MAP inference under a correlated prior, as desired. Note that the connectivity statistics play two roles: first, the mean strength $\overline{A}_{ij}$ of the connections between the unit representing feature $j$ and the sisters sampling receptor $i$ encodes the affinity of the receptor for that feature. Second, the prior correlation $C_{jk}$ relating two features is determined by the covariance of the connectivity between the sisters and the units representing those features. We show this schematically in Fig. 1A.

## 3 Connectivity for correlated priors

How do we encode a desired affinity and correlated prior into the connectivity? As we show below, we may not be able to encode correlations among all $N$ features. To achieve the desired affinities $\overline{A}_{ij}$ and correlated priors $C_{jk}$ on $n$ of the $N$ features, we will first assume that the affinities are zero, because once we have weights with the desired covariance, we can add the required affinities as an offset to all the weights, without affecting their covariances. The covariance of the weights that connect sisters sampling receptor $i$ to the units representing latent features $j$ and $k$ is then

$$C^i_{jk} = \frac{1}{S_i} \sum_s A^s_{ij} A^s_{ik}.$$

Our correlated priors set the weighted sum of covariances across input channels. So we must have

$$C_{jk} = \sum_i S_i C^i_{jk} = \sum_i \sum_s A^s_{ij} A^s_{ik}.$$

We see that correlated priors are encoded by the scalar product of the weights, when indexed by feature. To make this explicit we reshape the weights into $S$-element vectors, $\mathbf{a}_j$, one per feature $j$, making correlated priors scalar products of these vectors,

$$[\mathbf{a}_j]_{is} \triangleq A^s_{ij} \implies C_{jk} = \langle \mathbf{a}_j, \mathbf{a}_k \rangle = \sqrt{C_{jj} C_{kk}} \langle \hat{\mathbf{a}}_j, \hat{\mathbf{a}}_k \rangle = \sqrt{C_{jj} C_{kk}} \rho_{jk} \tag{9}$$

where $\rho_{jk}$ is the scalar product of the unit vectors, and we've used that $\|\mathbf{a}_j\| = \sqrt{C_{jj}}$ (see Fig. 1C).

The decomposition indicates that we can determine the weight vectors in two steps: First, we adjust their orientations to achieve the angles required by the scalar product. We do this by first ordering

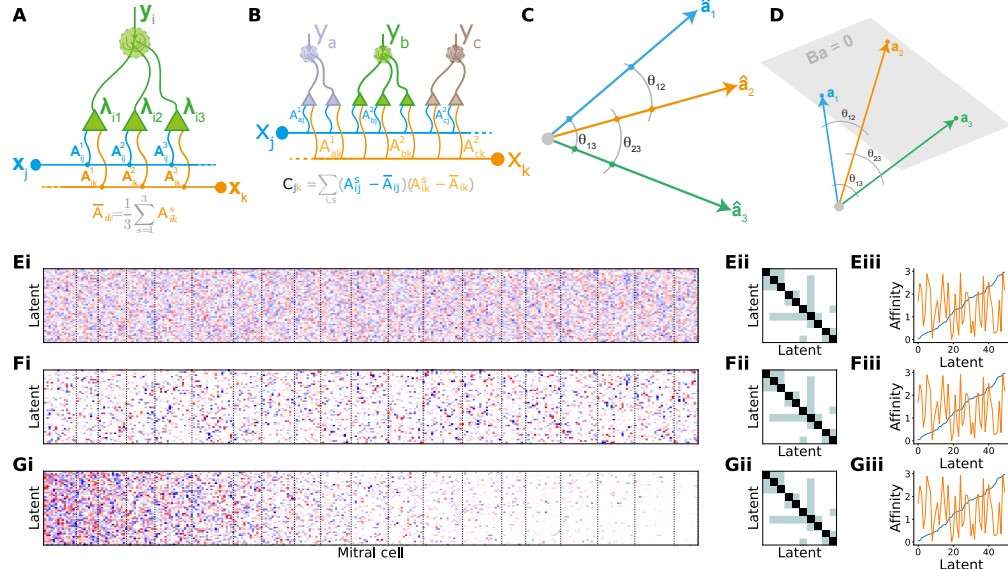

Figure 1: Encoding correlated priors with sister cells. **(A)** Schematic of the connectivity between sister cells and units representing latent features $j$ and $k$. The mean $\overline{A}_{ij}$ of the weights $A_{ij}^s$ connecting sister cells $\lambda_{is}$ sampling receptor $y_i$ to units $x_j$ and $x_k$ reflects the affinity of that receptor for those features. **(B)** The weighted sum of the covariances of the synaptic strengths with which units $x_j$ and $x_k$ connect to sisters sampling each input channel encodes the prior correlation between those features. **(C)** Desired correlations can be constructed by first assuming affinities are zero and adjusting the angles between vectorization $\mathbf{a}_j$ of the weights $A_{ij}^s$. After this adjustment the zero-affinity condition will likely be violated, but can be satisfied by **(D)** rotating the weights into the null space of the constraint matrix $\mathbf{B}$. Weights are then rescaled by the desired standard deviations, and the required affinities are added. **(Ei)** Example weight matrix connecting mitral cells (columns) and latent feature units (granule cells, rows). Neighbouring mitral cells are sisters except across input channel boundaries, indicated by vertical dotted lines. Weights are coloured by their deviation from the affinity of a given channel for a given latent. **(Eii)** Correlated prior achieved by the weight matrix in panel Ei, showing only the first 10 latents for clarity. **(Eiii)** Affinity of the first (blue) and second (orange) input channels for the latents, ordered by the former. **(Fi-Giii)** Many weight matrices can achieve the same correlated prior and affinity, for example the ones shown in panels Eii and Eiii. The remaining panels are as in Ei-Eiii, but for a weight matrix that was optimized for sparsity i.e. changing as few synapses as possible from the value dictated by the affinity **(Fi-Fiii)**, and one that was optimized for weighted sparsity, where some input channels had more changes than others **(Gi-Giii)**. See Sec. S2.1 for details.

the unit vectors arbitrarily, then taking the first and assigning it to be the first standard unit vector $\mathbf{e}_1 = [1, 0, \dots]$, the second vector to be $[\rho_{12}, \sqrt{1 - \rho_{12}}, 0, \dots]$ and so on. Continuing in this way we define unit vectors for all $n$ features since for the $k$'th weight vector, we use the first $k - 1$ elements to achieve the correlations with the previously considered features, the $k$'th element to achieve unit length, setting the remaining $n - k$ elements to zero. At the end of this procedure, the $n$ feature vectors are at the desired orientations relative to each other. We then simply scale each unit vector to its desired length, as specified by the elements $\sqrt{C_{jj}}$.

It is clear from this geometric formulation of the problem that we have a rotational degree of freedom: any fixed rotation applied to all of the feature vectors will retain their lengths and relative orientations, and therefore the desired co-occurrence data. We will use this rotational degree of freedom to achieve the desired zero-affinity condition. This condition states that $\sum_s A_{ij}^s = 0$ for all input channels $i$ and latent features $j$. To relate this sum to our weight-vectors we convert it to a sum over sisters $\underline{\text{and}}$ input channels by defining a binary indicator,

$$B_{im}^s = \begin{cases} 1 & \text{if } i, s \text{ indexes a sister that samples input channel } m, \\ 0 & \text{otherwise.} \end{cases}$$

The zero-affinity condition then becomes

$$\sum_i \sum_s A_{ij}^s B_{im}^s = 0 \quad \text{for all input channels } m \text{ and latent features } j.$$

By converting each $B_{im}^s$ into an $S$-element vector $\mathbf{b}_m$ like we did the weights, this condition becomes

$$\langle \mathbf{b}_m, \mathbf{a}_j \rangle = 0 \quad \text{for all input channels } m \text{ and features } j.$$

We can specify these $MN$ conditions in a single matrix equation by stacking the $M$ vectors $\mathbf{b}_j$ into the $M \times S$ matrix

$$\mathbf{B} \triangleq [\mathbf{b}_1, \mathbf{b}_2, \ldots \mathbf{b}_M]^T,$$

and the $N$ vectors $\mathbf{a}_j$ into the $S \times N$ matrix

$$\mathbf{W} \triangleq [\mathbf{a}_1, \mathbf{a}_2, \ldots \mathbf{a}_N],$$

whereby our zero-affinity condition becomes

$$\mathbf{BW} = \mathbf{0}.$$

To see that solutions exist, notice first that $\mathbf{B}$ has an $S - M$ dimensional nullspace. Notice also that $\mathbf{W}$ has an $n$-dimensional column space, because we only specified weight-vectors for $n$ features, and left the rest at zero. Therefore, as long as

$$n \leq S - M$$

we can always rotate $\mathbf{W}$ into the null-space of $\mathbf{B}$, and achieve the zero-affinity condition (see the schematic in Fig. 1D). This bound says, first, that we can only specify covariance priors if we have more sisters than input channels. Second, it says that the number of features for which we can encode correlated priors grows with the number of sister cells. The value $S - M$ counts how many more sisters we have than input channels. If this number is smaller than the number of latent features $N$ of interest to the system, it has to choose the most important $n$ for which to encode correlations.

To determine the set of all possible solutions, we decompose $\mathbf{W}$ into $\mathbf{U}_W \mathbf{S}_W \mathbf{V}_W^T$ using singular value decomposition (SVD), where we've taken $\mathbf{U}_W$ to be $S \times n$. Rotating this column space leaves the correlations $\mathbf{W}^T \mathbf{W}$ unchanged. To find the subset of rotations that satisfy the affinity condition $\mathbf{BW} = \mathbf{0}$, we apply SVD to $\mathbf{B}$ and get a basis for its row space in the columns of the $S \times M$ matrix $\mathbf{V}_B$. Letting the $S \times S - M$ matrix $\mathbf{V}_B^\perp$ be an arbitrary orthogonal completion of this basis, we see that the affinity condition is met if and only if $\mathbf{V}_B^T \mathbf{U}_W = 0$, i.e. if the span of $\mathbf{U}_W$ is in the span of $\mathbf{V}_B^\perp$. We can therefore choose the first column of $\mathbf{U}_W$ to be any weighted combination of the $S - M \triangleq m$ columns of $\mathbf{V}_B^\perp$ that has unit norm, giving $m - 1$ degrees of freedom. The second column of $\mathbf{U}_W$ can be selected in the same way, but orthogonal to the first, giving $m - 2$ degrees of freedom. Continuing in this way until we have selected all $n$ columns of $\mathbf{U}_W$, we see that we have $\sum_{i=1}^n (m - i) = nm - \frac{1}{2}n(n+1)$ degrees of freedom when determining $\mathbf{U}_W$.

In practice, we find solutions by first applying an $m$-dimensional rotation to $\mathbf{U}_W$, then picking its first $n$-columns. We can pick rotations randomly, or to optimize certain properties of the resulting weight matrices. In Fig. 1E-G we show three different weight matrices, all producing the same affinity and correlated prior, but with different sparsity properties. We investigate the effects of these difference on sister cell responses in Section 5 below.

## 4   Inference with correlated priors

If latent features are correlated, then inference that incorporates this information will outperform that which does not. To verify this, we considered the receptor input produced by the simultaneous presence of five latent features at high concentration, and corrupted by noise of a fixed variance. We compared the performance of two inference circuits. The first treated all latent features as independent. For the second, we used the approach in Sec. 3 to construct connectivity that encoded prior correlations on just the five features present. In Fig. 2A we compare the inferred feature concentrations at convergence for both circuits. It's clear that the circuit that incorporates the correlated prior outperforms the vanilla circuit.

In Fig. 2B we show the time course of activity in the granule cells encoding latent features for the circuit that uses the correlated prior, when presented with the input in panel A. The dynamics, which

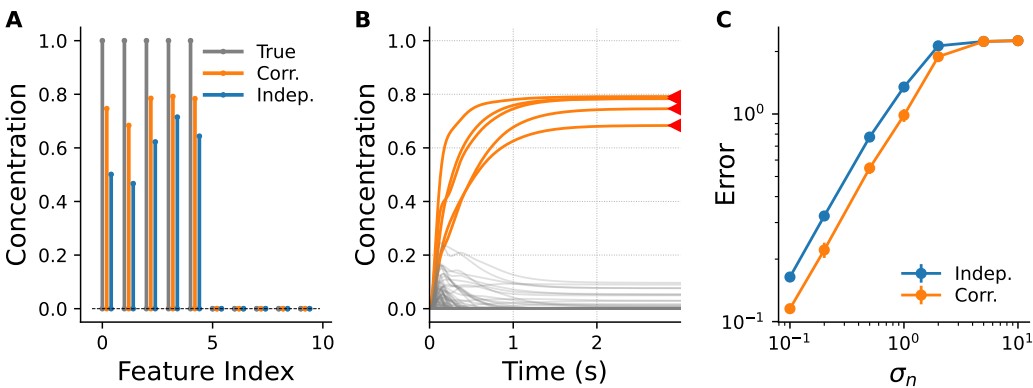

Figure 2: Inference with correlated priors. We simulated a network with 50 input channels, each with a uniform random number of 4-9 sisters, and 200 latent units (see Supplementary Information for additional details.) **(A)** Results of inferring the true feature values (gray) when using a correlated prior (orange), or assuming feature independence (blue), with receptor noise s.d. of 0.5. Only the first 10 of 200 features are shown, for clarity. **(B)** Time course of inference readout in the latent feature units, the first five of which (orange) correspond to the features actually present, for the correlated prior setting of panel A. Red triangles indicate the exact solution to the problem for those features as determined by convex optimization. **(C)** Mean (dots) +/- s.d. (bars) of the error (Euclidean norm) between the true and inferred feature vectors computed over 5 random noisy corruptions of the same receptor input, for different receptor noise standard deviations. See Sec. S2.2 for details.

use biologically realistic time constants, show that the inference result is achieved within a few 100 ms, consistent with the time-course of respiration.

Finally, we compared the performance of the two circuits over a range of receptor noise settings. An important parameter of the inference circuity is the assumed variance of the receptor noise, which can differ from the true noise level and can be adjusted to improve MAP inference. Therefore, for each level of input noise we reported the performance of the circuit using the inference noise that gave the lowest error. The results for both circuits are plotted in Fig. 2C and reveal that the circuit using correlated priors outperforms the vanilla circuit at all but the highest noise levels, where both circuits perform equally poorly.

## 5 Effect of different connectivity solutions

Next, we examine the effect of connectivity solutions on response heterogeneity. We saw in Section 3 that many connectivity solutions exist that produce a desired correlated prior and affinity. Different connectivity solutions will produce different levels of heterogeneity in sister cell responses. For example, sparse solutions like those in Fig. 1F, in which differences in the connectivity of sisters are limited to a few sites, will result in more homogeneous responses, while dense solutions, like those in Fig. 1E where sister cell connectivity varies widely, will produce more heterogeneous responses. The heterogeneity of responses can therefore inform about the connectivity structure of the circuit, as we demonstrate by comparing to experimentally recordings from the olfactory system.

In Fig. 3Ai we show the responses of the first (green) and second (olive) sister cells sampling the first input channel to three different stimuli presented at fixed concentration for one second starting at $t = 0$ (gray), for the system using the random connectivity of Fig. 1E. We see that the responses to some stimuli e.g. stimulus 2, are quite similar, while those to others, e.g stimulus 1, can be different. In Fig. 3Aii we summarize this response heterogeneity by computing the Pearson correlation of responses over the t = 0-1.5 sec time window and averaging over all pairs of sisters in each input channel, yielding one value per channel and stimulus. We took values less than 0.3 (brown) as indicating that the sisters in an input channel had diverse responses to that stimulus, while values above 0.7 (bronze) we took to indicate stereotyped responses. In Fig. 3Aiii we show how response heterogeneity was distributed per odour channel, ordered by the average similarity of responses. Since

the random connectivity used was distributed evenly among input channels, we see that response heterogeneity is similar across all channels.

In Fig. 3Bi-Biii we perform the same procedure as in panels Ai-Aiii, but for a system that used the uniform sparse connectivity of Fig. 1F. Because differences in the connectivity of sisters cells to latent feature units occurred at fewer sites, sister cell responses are more homogeneous, and similar across all input channels. Finally, in Fig. 3Ci-Ciii we do the same as in the previous panels, but now using the weighted sparse connectivity of Fig. 1G. In this setting, input channels with lower indices receive more connectivity changes. This results in high diversity of responses in those channels, as seen in the individual responses of panel Ci and the summary statistics of panel Ciii. The higher density of changes for sisters sampling some input channels, compared to the low density in others also results in larger fractions of both diverse, and stereotyped, responses, as revealed by panel Cii.

The patterns of response heterogeneity can suggest the connectivity of a given circuit. We demonstrate this by comparing our simulated responses in panels A-C, to those recorded in the olfactory system by [9]. In Fig. 3Di we show experimentally recorded odour responses of pairs of sisters sampling three different input channels, revealing that in all three cases, sisters can respond similarly to some odours, but differently to others, like we saw in simulations. In Fig. 3Dii we show the distribution of response heterogeneity per olfactory input channel. The prominent presence of both diverse and stereotyped responses, and their uneven distribution among input channels, is similar to our simulations using a weighted sparse connectivity in panels Cii, suggesting that similar connectivity in the recorded olfactory system. However, the overall distribution of responses for the experimental data, shown in Fig. 3Diii shows more diverse responses, and a different trend in the distribution than for our simulated data of Fig. 3Ciii. We address possible reasons for this in the Discussion.

## 6   Estimating correlated priors from responses

Inference using priors that reflect natural feature statistics would improve inferential accuracy. Therefore, natural and artificial systems performing sensory inference would be predicted to use such priors. In our theory, these priors are encoded in the strengths of connections between sister cells and latent feature units (granule cells). Directly measuring these strengths is difficult, while sister cell responses are much easier to measure. Can we infer the priors from sister cell responses alone?

The principal difficulty in determining connectivity from sister cell responses is that multiple latent feature units may respond to a given stimulus. Therefore, the steady-state response of a sister cell reflects the influence of a corresponding number of weights, making it difficult to determine the strength of any particular one.

To avoid this problem, we appeal to the assumed sparsity of feature representations and assume a 'one-hot', or 'grandmother cell' model of latent feature unit responses in which only one is activated per stimulus. Thus when stimulus $j$ is presented, the feature unit activation vector at the end of inference is approximately,
$$\hat{\mathbf{x}}[j] = [0, 0, 0..., 0, c_j, 0, 0, 0, ...0],$$
where $c_j$ is the inferred value of the stimulus, and we've used $[j]$ to emphasize that this is the response to stimulus $j$. From Eqn. (4) the corresponding steady-state activity of sisters cells simplifies to
$$\sigma^2 \lambda_{is}[j] = y_i[j] - A_{ij}^s c_j.$$

We still need the input $y_i$ to this sister cell, but this is common to all sister cells sampling input channel $i$, and is eliminated in the covariance computation (see below).

We can rearrange the steady-state response and express the sister weights in terms of the activity as
$$A_{ij}^s = (y_i[j] - \sigma^2 \lambda_{is}[j])/c_j.$$

To compute the covariance we also need the average value of the weights. Since all sisters receive the same input $y_i$, this average is simply
$$\overline{A}_{ij} = (y_i[j] - \sigma^2 \overline{\lambda}_i[j])/c_j,$$

so the difference we need is
$$A_{ij}^s - \overline{A}_{ij} = -\sigma^2 (\lambda_{is}[j] - \overline{\lambda}_i[j])/c_j.$$

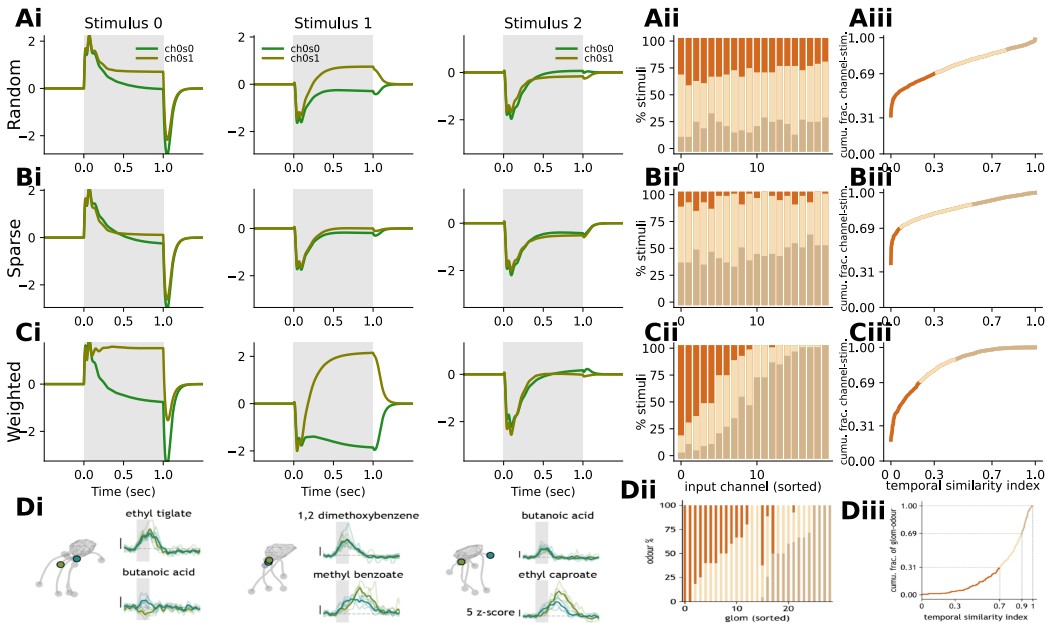

Figure 3: Effect of different connectivity solutions on inference. **(Ai)** Activity of the first (green) and second (olive) sister cells in the first input channel to three different stimuli, presented at fixed concentration during the gray time window, when using the random connectivity of Fig. 1E. **(Aii)** Distribution of sister cell response heterogeneity, computed as the Pearson correlation of responses of two sisters from t=0 to t=1.5 seconds, averaged over all pairs of sisters in each input channel, yielding one value for each input channel-odour pair. Values below 0.3 (brown) are termed diverse, those above 0.7 (bronze) stereotyped. **(Aiii)** Distribution of response types per input channel, sorted by prevalence of stereotyped responses. **(Bi-iii.)** As in panels Ai-iii but for a system using the sparse connectivity of Fig. 1F. **(Ci-iii)**. As in panels Ai-iii but for a system using the weighted sparse connectivity of Fig. 1G. **(Di)** Calcium responses (traces) from pairs of experimentally recorded sisters cells from three different olfactory input channels to two different odours per channel [9]. **(Dii,iii)** As in Aii-iii but for the experimentally recorded responses. See Sec. S2.3 for details.

In other words, under our sparsity assumption about latent features, the deviation of the weights from their average is proportional to the deviation of the sister cell activities from their average.

We can now use sister cell responses to compute the contribution of the $i$'th input channel to the prior,

$$C^i_{jk} = \frac{1}{S_i} \sum_s (A^s_{ij} - \overline{A}_{ij})(A^s_{ik} - \overline{A}_{ik})$$

$$= \frac{\sigma^4}{S_i} \sum_s \frac{\lambda_{is}[j] - \overline{\lambda}_i[j]}{c_j} \frac{\lambda_{is}[k] - \overline{\lambda}_i[k]}{c_k}$$

$$= \frac{\sigma^4}{c_j c_k} \operatorname{cov}_s(\lambda_i)_{jk}.$$

The correlation prior is the weighted sum of these

$$C_{jk} = \sum_i S_i C^i_{jk} = \frac{\sigma^4}{c_j c_k} \sum_i S_i \operatorname{cov}_s(\lambda_i)_{jk}. \tag{10}$$

Therefore, to determine the correlated priors from the sister cell activations, we also need to know the inferred values of each stimulus. We can address this requirement in several ways.

One possibility is to assume that all the inferred stimulus values are the same, $c$. In that case,

$$C^i_{jk} = \sigma^4 c^{-2} \operatorname{cov}_s(\lambda_i)_{jk},$$

so, the correlated prior is proportional to the variance of the sister cell activations. In Fig. 4A we have plotted the correlated priors estimated in this way from the experimentally recorded sister cell responses [9] (see Sec. S2.4). The results suggest a broad trend of mild anticorrelation. There is positive correlation of methyl valerate, which has a fruity odour [10], with, for example with 2-heptanone, which has 'banana-like, fruity odour' [11]. This can suggest a prior modelling the co-occurrence of fruity odours. Nevertheless, the most prominent feature of the panel is the block of strong negative correlations, which include that of methyl varelate with valeraldehyde, which has 'fruity, nutty, berry' odour [12], contradicting a simple association of fruity odours.

The counterintuitive priors in Fig. 4A were computed assuming constant inferred odour concentrations. Another possibility is that these concentrations are related to the vapour pressure (see Table S1). Because the vapour pressures varied widely, using them directly resulted in large fluctuations in the estimated priors. Instead, we selected a function that produced visually smoother estimates, whereby

$$c_j^{-1} = 2 + \frac{1}{2(v_j + 0.1)},$$

where $v_j$ is the vapour pressure of that odour. In Fig. 4B shows this approach has reduced the block of anticorrelation around methyl valerate. This procedure suggests how concentration functions can be fit to natural statistics. In fact, when sufficient data is present, the inferred concentrations can be treated as free parameters and fit individually to any stimulus correlation data. Note, however, that because concentrations are positive, adjusting them can only change the magnitude of an inferred prior correlation, not its sign – see Eqn. (10).

A final possibility that we considered was to assume that because correlated priors can only be specified between a subset of all possible pairs of odours, the priors that are encoded are likely to be strong. We can then find inferred concentrations that maximize the magnitude $|C_{jk}|$ of the encoded priors. We first defined the weighted sum of activity covariances,

$$Q_{jk} \triangleq \sum_i S_i \, \text{cov}_s(\lambda_i)_{jk}.$$

The magnitude of the correlated prior can then be written as

$$|C_{jk}| = \sum_i S_i |C_{jk}^i| = \frac{\sigma^2}{c_j c_k} \left| \sum_i S_i \, \text{cov}_s(\lambda_i)_{jk} \right| = \frac{\sigma^2}{c_j c_k} |Q_{jk}|.$$

We then searched for the inferred concentrations that maximized the summed magnitude of the correlation priors. Because inferred concentrations appear in the denominator, we needed to avoid degenerate solutions that drive them to zero. So rather than working with concentrations, we used inverse concentrations $r_j \triangleq \frac{1}{c_j}$. We then found the vector $\mathbf{r} \triangleq [r_1, \ldots, r_n]$ of inverse inferred concentrations that maximized the summed magnitude of the correlated priors. That is, we solved

$$\underset{\|\mathbf{r}\| \le 1}{\text{argmax}} \sum_{j,k} |C_{jk}| = \underset{\|\mathbf{r}\| \le 1}{\text{argmax}} \sum_{j,k} \sigma^2 r_j r_k |Q_{jk}|,$$

where the length constraint on $\mathbf{r}$ avoids degenerate maximization by scaling of the inverse concentrations. Given this formulation, the solution to the above problem was the principal eigenvector of $|Q_{jk}|$. In Fig. 4C we have plotted the correlated prior estimated in this way. By construction, many more terms have elevated values, maximizing the sum of absolute correlations.

To quantitatively evaluate the quality of these estimates, we consulted a publicly available database of 214 essential oils and their monomolecular components [13]. A subset of these monomolecular odours were also used in the experiments of [9] - these are indicated by the dotted lines the pannels of Fig. 4. We then searched the essential oils database and marked any pairs that co-occurred in at least one essential oil with a black rectangle. Assuming that the essential oils, which are extracted from plants, are ethologically relevant enough for the animals to encode co-occurrence information about them, we would expect the correlated priors to reflect this, and the corresponding pairs to be postive (green) in the panels of Fig. 4. What we we actually observe is that the estimated terms are all near zero, suggesting lack of correlation. We comment on this observation in the Discussion.

## 7 Discussion

In this work we have taken inspiration from the olfactory system to show how natural and artificial systems performing inference can use sister cells – units that receive the same input but connect

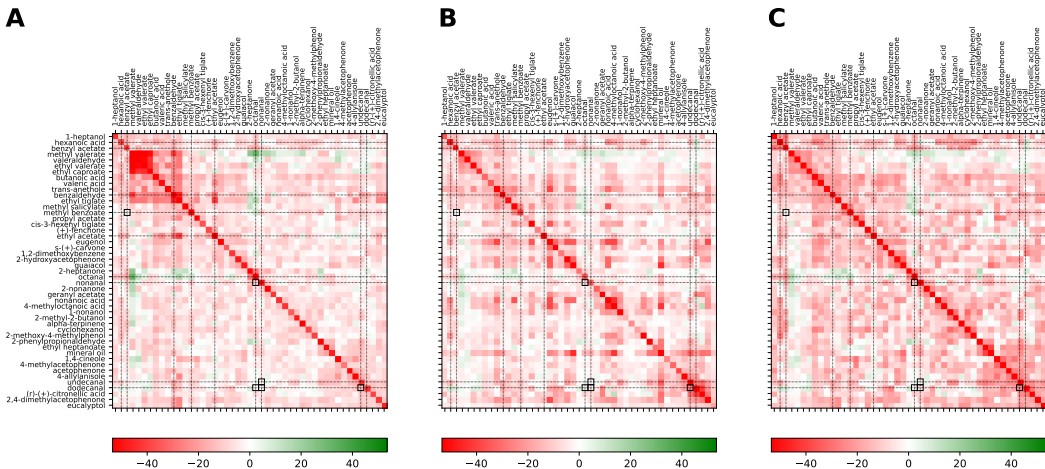

Figure 4: Estimating correlated priors from the experimentally recorded sister cell responses in [9]. The values shown are $-C_{jk}$, so positive (green) means a prior promoting correlation, negative (red) is anti-correlation. Estimates when assuming inferred concentrations are **(A)** are constant, **(B)** a fixed function of the vapour pressure of each odour, or **(C)** determined by the eigenvectors of the response covariance. Dotted lines mark odours present in the essential oils database of [13], squares indicate odour pairs that co-occur in at least one essential oil in that database. See Sec. S2.4 for details.

differently to units representing latent variables – to incorporate correlated priors on the latents, without requiring direct interactions between the latent units. We used geometric arguments to show how the connectivity between sister cells and the latent units can be constructed, and verified improved inference performance when latents were correlated. We demonstrated how different connectivity solutions can affect the heterogeneity of sister cell responses, providing clues about connectivity from responses alone. Finally, we showed how under certain assumptions about latent representations, the correlated priors used by a system can be estimated from the sister cell responses alone. Although our approach is derived from the olfactory system, the ideas involved are general and should be applicable to other natural and artificial systems that perform inference in environments with correlated latents.

**Limitations.** Our work is a simple proof-of-concept and has a number of limitations. A key aspect of our approach is the linear, isotropic Gaussian observation model of receptor responses, which lends itself to completing the square and from which the correlated prior emerges. In many systems, including the olfactory system from which we take inspiration, such a model may be inappropriate or too simplistic, and it is unclear whether extending more realistic models to use sisters cells would readily yield correlated priors. Further, more testing than the single stimulus corrupted by a range of noise that we used in Section Sec. 4 would be needed to robustly establish the performance of the model. In Section 3 we used geometric arguments to demonstrate how to find connectivity that achieves a desired stimulus affinity and correlated prior. An important extension of this work would be to show how natural and artificial systems can learn such connectivity from natural stimulus statistics. In that section we also showed how a variety of connectivity solutions exist and in Section 5 we explored the effects of different solutions on the heterogeneity of sister cell responses, and compared them to those observed in the olfactory system. The weighted sparse connectivity, in which sisters in some channels had more heterogeneous synaptic strengths than others, qualitatively matched the per-channel heterogeneity statistics (compare Fig. 1Cii,Dii), but fell short on the pooled statistics (Fig. 1Ciii,Diii). However, we did not directly optimize the sparsity weighting to match these statistics, and doing so may improve the match further, and suggest a similar connectivity in the olfactory bulb. In Section 6 we showed how priors can be estimated from sister cell responses alone. This was only possible because of our assumption of 'grandmother-cell' latent feature representations. We have not explored whether estimation is possible when this assumption is relaxed.

## Acknowledgements

This work was supported by the Francis Crick Institute, which receives its core funding from Cancer Research UK (CC2036 to A.T.S.), the UK Medical Research Council (CC2036 to A.T.S.), and the Wellcome Trust (CC2036 to A.T.S.). This work was also supported by the National Science Foundation / Canadian Institutes of Health Research / German Research Foundation / Fonds de Recherche du Quebec / UK Research and Innovation–Medical Research Council Next Generation Networks for Neuroscience Program (Award No. 2014217 to A.T.S.). We are grateful to the members of the Schaefer lab at the Francis Crick Institute and the Latham lab at the UCL Gatsby Computational Neuroscience Unit for useful discussions.

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

## Supplementary Information

### S1    Derivations

To show that the fixed points of the dynamics in Eqn. (4) and Eqn. (5) minimize the loss in Eqn. (6), we set the temporal derivatives to zero. Doing so for Eqn. (4) gives

$$\sigma^2 \lambda_{is} = y_i - \sum_j A_{ij}^s x_j.$$

Doing so for Eqn. (5) and using the above expression for $\lambda_{is}$ gives

$$\phi_j'(x_j) = \frac{1}{\sigma^2} \sum_i \sum_s A_{ij}^s (y_i - \sum_{ij}^s A_{ij}^s x_j).$$

Setting the partial derivative of the loss in Eqn. (6) with respect to $x_j$ gives the same equation, proving that the fixed points of the dynamics are the same as those of the loss. The latter will be minima when the $\phi_j(x_j)$ are the convex functions typically used in the literature.

To show that the loss in Eqn. (8) is equivalent to that in Eqn. (6), let's focus on the contributions to the former loss from a single input channel. Expanding it out and combining terms over sisters,

$$\sum_s (y_i - \sum_j A_{ij}^s x_j)^2 = S_i y_i^2 - 2y_i \sum_j \left( \sum_s A_{ij}^s \right) x_j + \sum_j \sum_k \left( \sum_s A_{ij}^s A_{ik}^s \right) x_j x_k.$$

The first term is $S_i$ copies of the squared input $y_i^2$, since each sister will receive the same input. The second term involves a term summing the weights $A_{ij}^s$ connecting sister cells sampling input channel $i$ to granule cell $j$. We can represent this sum as the average of the weights $\overline{A}_{ij}$, scaled by the number of sisters $S_i$ sampling that input channel. We then have

$$\sum_s (y_i - \sum_j A_{ij}^s x_j)^2 = S_i \left( y_i^2 - 2y_i \sum_j \overline{A}_{ij} x_j \right) + \sum_j \sum_k \left( \sum_s A_{ij}^s A_{ik}^s \right) x_j x_k.$$

The first term in brackets is the contribution to the loss from an input channel with a single mitral cell but is missing the pairwise interaction terms between the granule cells. We can add this missing term, but must also subtract it to leave the overall sum unchanged. 'Completing the square' in this way, we arrive at

$$\left( y_i^2 - 2y_i \sum_j \overline{A}_{ij} x_j \right) + \sum_j \sum_k \overline{A}_{ij} \overline{A}_{ik} x_j x_k - \sum_j \sum_k \overline{A}_{ij} \overline{A}_{ik} x_j x_k = (y_i - \sum_j \overline{A}_{ij} x_j)^2 - \sum_j \sum_k \overline{A}_{ij} \overline{A}_{ik} x_j x_k.$$

Substituting this into our expression above,

$$\sum_s (y_i - \sum_j A_{ij}^s x_j)^2 = S_i (y_i - \sum_j \overline{A}_{ij} x_j)^2 + \sum_j \sum_k \left( \sum_s A_{ij}^s A_{ik}^s \right) x_j x_k - S_i \sum_j \sum_k \overline{A}_{ij} \overline{A}_{ik} x_j x_k.$$

Factoring out $S_i$ from the middle term and combining interaction terms, we get

$$\sum_s (y_i - \sum_j A_{ij}^s x_j)^2 = S_i (y_i - \sum_j \overline{A}_{ij} x_j)^2 + S_i \sum_j \sum_k \left( \frac{1}{S_i} \sum_s A_{ij}^s A_{ik}^s - \overline{A}_{ij} \overline{A}_{ik} \right) x_j x_k.$$

We recognize the last term in brackets as the covariance $C_{jk}^i$ of the weights with which sister cells sampling input channel $i$ connect to granule cell $j$ and granule cell $k$. We can finally express the contribution to the loss from input channel $i$ as

$$\sum_s (y_i - \sum_j A_{ij}^s x_j)^2 = S_i (y_i - \sum_j \overline{A}_{ij} x_j)^2 + S_i \sum_j \sum_k C_{jk}^i x_j x_k.$$

Returning to the loss in Eqn. (6) we now see that we can write it as

$$\mathcal{L}(\mathbf{x}) = \sum_j \phi(x_j) + \sum_i \frac{S_i}{2\sigma^2} (y_i - \sum_j \overline{A}_{ij} x_j)^2 + \frac{1}{2\sigma^2} \sum_j \sum_k \sum_i S_i C_{jk}^i x_j x_k^i,$$

which after pooling covariance terms across input channels, yields the loss in Eqn. (8).

## S2   Simulation Details

The standard deviation of the receptor noise and that of inference were $\sigma_n = 0.5$ and $\sigma_{\text{inf}} = 20$, respectively, unless otherwise noted. In all simulations the individual prior on the latents was the elastic net

$$\phi_i(x_i) = \beta x_i + \frac{\gamma_i}{2} x_i^2,$$

where the $\ell_1$ parameter $\beta = 0.1$, unless otherwise noted, and the $\ell_2$ parameter $\gamma_i$ was set per unit so that its sum with the corresponding diagonal term coming from the correlated prior was 0.1. These values for the parameters of the loss function were selected because they gave good inference performance for the example inputs used in the text. The integration time constants of the mitral cells and latent feature units were $\tau_{\text{mc}} = 50$ msec. and $\tau_{\text{gc}} = 100$ msec., respectively. These were selected because they were biologically realistic and gave smooth dynamics that converged within respiration time. We used first-order Euler integration with a step size of $200\mu\text{sec}$ to integrate the dynamics. All simulations were carried out in python version $\geq 3.9$ running on a mid-2015 2.8 GHz Intel Core i7 MacBook Pro, and all individual simulations ran in about one minute or less.

### S2.1   Figure 1

To generate the connectivity matrices in Fig. 1E-G we first generated our desired affinity and correlated prior. We simulated a system with $M = 20$ input channels and $N = 50$ latents. The affinities of the input channels for the latents were selected independently from the uniform distribution over $[0 - 3]$. The number of sister cells sampling each input channel was selected independently and uniformly at random from the integer range $10 - 20$. Given this large number of sister cells per input channel, we could use a correlated prior that involved all $N = 50$ of the latents. We generated such a prior by setting a random 10% of the upper triangular elements of an $N \times N$ matrix to 0.1, adding the result to its transpose, and finally setting the diagonal elements to 1. The sparsity and strength of the correlations were set to ensure that the resulting matrix remained positive definite. We then scaled the result by $\sigma_{\text{inf}}^2 \gamma$, so that when scaled down by $\sigma_{\text{inf}}$ (see Eqn. (8)) the diagonal elements would equal the $\ell_2$ prior $\gamma$.

With the affinity and correlated prior selected in this way, we generated connectivity to achieve it using the geometric approach described in Sec. 3. We first generated the $S \times N$ matrix $\mathbf{W}$ with unit norm columns and angles set by the desired correlations, and performed SVD on it, keeping the product of the singular values and right eigenvectors $\mathbf{S}_W \mathbf{V}_W^T$. We then constructed the constraint matrix $\mathbf{B}$, computed its right eigenvectors $\mathbf{V}_B$ and an orthonormal complement $\mathbf{V}_B^\perp$. We replaced the column space of $\mathbf{W}$ by applying a random $N \times N$ orthonormal matrix $\mathbf{R}$ to this complement, so (updating $\mathbf{W}$),

$$\mathbf{W} = \mathbf{V}_B^\perp \mathbf{R} \mathbf{S}_W \mathbf{V}_W^T.$$

The matrix in Fig. 1Ei used such a random matrix. For the weight matrix in Fig. 1Fi we optimized over $\mathbf{R}$ to find sparse solutions by minimizing the sum $\beta_W |\mathbf{W}|$ with $\beta_W = 10$, doing so 5 times with random initializations and keeping the sparsest result. For the weight matrix in Fig. 1Gi we used a weighted penalty based on input channel, where $\beta_W$ for weights for sister cells sampling the first channel was 0, those sampling the second was 1, and so on. We again repeated this procedure 5 times with random initialization, and kept the best result. Once $\mathbf{W}$ was determined in this was, we scaled the columns of $\mathbf{W}$ to their required lengths. Optimizations were performed using the `pymanopt` package using its SteepestDescent optimizer. The resulting matrices are what are plotted in Fig. 1. In the final step, we added the affinities of the corresponding channel for the corresponding latent to every weight, to achieve the affinity condition.

### S2.2   Figure 2

To generate the inference results in Fig. 2A we simulated a system with $M = 50$ input channels, $N = 200$ latents, and a random number between 4 to 9 sisters per channel. Our correlated prior promoted coactivation of the first $n = 5$ latents. We achieved this with a correlation matrix with identity diagonal, and $-0.24$ off-diagonal for the first $n \times n$ block. This was then scaled by $-\sigma_{\text{inf}}^2 \gamma$ (see previous section). Other parameters were as in Figure 1. In panel A we were only interested in the end result of inference so instead of simulating the dynamics we simply computed their solution by solving the convex optimization directly, using the `cvxpy` python package with the SCS solver. In Fig. 2B we were interested in the dynamics so simulated them, with the same parameters. In Fig. 2C we were

again only interested in the end result so we minimized the loss directly using convex optimization. We computed the results for different settings of receptor and inference noise standard deviation, testing every combination of $\sigma_n = \{0.1, 0.2, 0.5, 1, 2, 5, 10\}$ and $\sigma_{\text{inf}} = \{1, 2, 5, 10, 20, 50, 100\}$. The receptor standard deviation was used to generating random noise on each of 5 trials to corrupt the receptor input generated by the presence of the first $n$ features at unit concentration. For each circuit, we found the setting of $\sigma_{\text{inf}}$ that gave the best trial-averaged error at each noise level, and reported the results for each $\sigma_n$ using that setting.

## S2.3   Figure 3

The simulations in Fig. 3Ai,Bi,Ci we used the connectivity and parameters described in the details of Figure 1, and ran the dynamics of these three circuits for three stimuli, each consisting of a single latent at unit concentration. Temporal similarity indices for comparing sister cell responses were defined as the Pearson correlation of responses over the first 1.5 seconds following stimulus onset. In Fig. 3Aii,Bii,Cii we ran the three circuits for 50 stimuli, each consisting of one of the latent features at unit concentration. For each input channel and stimulus we then computed the average response similarity among all pairs of sister cells, and plotted these per channel, labeling similarities below 0.3 as diverse (orange), those above 0.7 as stereotyped (bronze). In Fig. 3Aiii,Biii,Ciii we plotted the cumulative distribution of response similarities pooled over all input channels and stimuli.

## S2.4   Figure 4

In Fig. 4 we estimated correlation priors from the experimentally recorded responses of [9] by assuming 'grandmother cell' feature representations to relate responses directly to connectivity. The only unknowns were then the inferred concentration for each odour. In Fig. 4A we assumed all these concentrations were the same constant value, in Fig. 4B we used an ad-hoc function of the vapour pressures listed in Table S1 and in Fig. 4C we used concentrations whose inverses, as a vector, was the principal eigenvector of the matrix of absolute value of covariances. Since the experimental data were pooled across multiple experiments, we first normalized the odour responses of each sister cell by its standard deviation across stimuli. We then scaled the odour responses by the inverse of the assumed inferred concentration for each odour. Next, for each input channel, we computed the biased covariance of the odour responses across sister cells. We used biased covariance so that the required division was by the number of sister cells, not this number mminus 1. We then weighted the resulting odour x odour covariance by the number of sisters in a glomeruls, and summed the result to produce the overall correlation prior (see Eqn. (10)).

## S3 Miscellaneous

Table S1: Vapour pressures for odours used in [9].

| Odour | Vapour Pressure (mmHg) |
|---|---|
| Nonanoic Acid | 9 |
| 2-Hydroxyacetophenone | 86 |
| 1-Nonanol | 41 |
| 2-Phenylpropionaldehyde | 294 |
| 1,2-Dimethoxybenzene | 0.47 |
| Ethyl Valerate | 4.745 |
| Trans-Anethole | 69 |
| 2-Nonanone | 645 |
| 2-Methyl-2-Butanol | 16.8 |
| Benzaldehyde | 1.27 |
| Hexanoic Acid | 158 |
| Methyl Valerate | 11.043 |
| Benzyl Acetate | 177 |
| 1-Heptanol | 325 |
| alpha-Terpinene | 1.64 |
| Acetophenone | 397 |
| Valeraldehyde | 31.792 |
| Geranyl Acetate | 256 |
| (+)-Fenchone | 463 |
| Ethyl Heptanoate | 0.68 |
| 4-Allylanisole | 0.21 |
| Cyclohexanol | 975 |
| Dodecanal | 34 |
| Propyl Acetate | 35.223 |
| 1,4-Cineole | 1.93 |
| Guaiacol | 78 |
| Butanoic Acid | 1.65 |
| Methyl Salicylate | 343 |
| 2-Methoxy-4-Methylphenol | 78 |
| 2-Heptanone | 4.732 |
| Nonanal | 532 |
| cis-3-Hexenyl Tiglate | 306 |
| Ethyl Caproate | 1.665 |
| Eugenol | 104 |
| S-(+)-Carvone | 66 |
| Methyl Benzoate | 0.38 |
| Octanal | 2.068 |
| Valeric Acid | 452 |
| Mineral Oil | 0 |
| 2,4-Dimethylacetophenone | 63 |
| Eucalyptol | 1.9 |
| Ethyl Tiglate | 4.269 |
| Undecanal | 83 |
| R-Citronellic Acid | 5 |
| Methyl Butyrate | 35.9 |
| R-(+)-Limonene | 198 |
| Ethyl Butyrate | 12.8 |
| 4-Methyloctanoic Acid | 6 |
| Ethyl Acetate | 111.716 |
| 4-Methylacetophenone | 187 |

