# OpenReview forum: "Inference with correlated priors using sisters cells"
_NeurIPS.cc/2025/Conference — NeurIPS 2025 poster_

### Official Review · Reviewer_aGi5 · 2025-06-11

**Clarity:** 2
**Significance:** 2
**Originality:** 3
**Rating:** 3
**Confidence:** 2

**Summary:**

The paper proposes a biologically-plausible formulation that lets a neural network perform infenence with correlated latent variables, even when the neurons representing those latents are not connected directly to one another.

On simulated data, the correlated prior performed better than it's uncorrelated counterpart.

Some theoretical arguments where made about dense random weights vs. weighted sparse patterns, which best matched experimental calcium responses, and serve to act as a connection between the method and empirical neurophysiological results.

**Questions:**

- What's the notable take-home message from this paper?
- Is the novelty here trying to test a neuroscientific theory in-silico, or is it to use a biologically-inspired mechanism to introduce a new algorithm?

**Ethical Concerns:**

["NO or VERY MINOR ethics concerns only"]

**Final Justification:**

Work has potential, but to make the right impact, some utility on real data needs to be illustrated. For example a neuroscience dataset.

**Limitations:**

I appreciate the authors acknowledgment of weaknesses, which seems to echo my point about the findings in this paper being unclear and potentially not self-contained in-terms of a final message/insight. Given the above, what's the real take-home message of this paper?

**Paper Formatting Concerns:**

Did not notice any.

**Quality:**

3

**Strengths And Weaknesses:**

**Strengths**
- Creative method, with a relatively thorough attempt to mathematically motivate the work.
- Attempt to close the loop between empirical observations and algorithmic inspiration. I appreciate the attempt to motivate the method through a neuroscientific lens, and then subsequently attempt to link the results back to the observed data.

**Weaknesses**
- Only applied to hand-designed, simple data. The significance of the results is very unclear to me. Especially as it pertains to modern ML methods.
- The connection between the neuroscientific insight and algorithm is weak
- The significance of the results in general are unclear and open to interpretation.

---

> ### Author Rebuttal · Authors · 2025-07-30
>
> We thank the reviewer for their comments and apologise for any lack of clarity in our presentation.
>
> The reviewer's first question, and their comment in the Limitations section, asks what the main message of our paper is. It is this: Many natural and artificial systems must perform inference about latents that are a priori correlated. The 'natural' way to incorporate such correlations requires direct interactions among the latents. This can be computationally or architecturally expensive when there are many latents. Our main contribution takes inspiration from the mammalian olfactory system to show an alternative approach to incorporating these correlations, by introducing 'sister cells' which receive the same input but connect differently to the latent units. This both provides a hypothesis as to why there are sister cells, and provides a new way of efficiently implementing inference for more realistic (correlated) priors
>
> The reviewer's second question is whether the novelty is testing a neuroscientific theory or proposing a new biologically-inspired algorithm. Our main aim is the latter: to propose a biologically inspired algorithm. Although we believe our algorithm sheds light on the structure and dynamics of the mammalian olfactory system, we believe it applies to other natural and artificial systems that perform inference of high-dimensional, a priori correlated latents.

---

> ### Comment · Reviewer_aGi5 · 2025-08-06
>
> Thank you to the reviewers for responding, alas I wished there were some additional experiments to prove the merit of their method. The state of AI right now is such that, that I think if you're proposing to propose a new algorithm, you can't just do it on toy data. You can at least train smaller equivalent models on math or other more tangible tasks. Look at the "Hierarchical Reasoning Model" papers as an example. I'm not asking for you to compete with ChatGPT, but if you're proposing a new learning algorithm, you **have to show empirical results on real data/benchmarks that people care about.**
>
> The authors made no attempt to introduce new experiments in any of their responses to the authors to sway my opinion, and as such I will recommend a rejection of the paper.

---

> > ### Comment · Reviewer_j9ec · 2025-08-06
> > **Their task is approximate Bayesian inference with correlated priors**
> >
> > There might be a benchmark model for that in posterior_db, but it doesn't really make sense to demand that they train with "real data/benchmarks that people care about" on a task more specific than "training models".

---

> > ### Author Response · Authors · 2025-08-06
> >
> > We thank the reviewer for emphasising the importance of empirical validation in ML research, particularly given the numerous new algorithms being proposed every day. We also thank them for pointing us to the impressive "Hierarchical Reasoning Model", which illustrates their point of testing new algorithms on important benchmarks that people care about.
> >
> > However, we point out that the aim and nature of our contribution differ from those of models like the HRM. Rather than proposing a new algorithm for end-to-end performance of important high-level cognitive tasks like Sudoku, as the HRM does, we show how one potential component of such computations, namely the incorporation of priors with correlation structure, can be achieved in a new way by using sister cells. Secondly, the HRM employs a variety of techniques such as fast and slow modules, approximate gradients, deep supervision, and adaptive computational time, whose cohesion and effectiveness are not immediately clear and therefore require empirical validation. In contrast, our contribution is based principally on the mathematical equivalence between the sister-cell loss (eq 6) and the loss incorporating correlated priors (eq 8). Establishing this equivalence requires mathematical derivations, which we provide, rather than numerical experimentation.
> >
> > Thus, we believe the nature of our contribution is such that its performance on benchmarks is of lesser importance than its demonstration of a new approach to incorporating correlated priors, which, to our knowledge, was previously unnoticed by the ML community. We believe it is a useful addition to the toolkit of ML researchers and therefore merits their attention. That is why we chose to focus our submission on the mathematical properties of the algorithm, rather than extensive testing on real-life datasets. However, we believe that our algorithm also provides insights into the structure and dynamics of the mammalian olfactory system. In future work establishing this correspondence, we will certainly follow the reviewer’s advice and test against the relevant datasets in that domain.

---

> > > ### Comment · Reviewer_aGi5 · 2025-08-08
> > > **Work has potential, but to make the right impact, it has to be improved.**
> > >
> > > I am not proposing rejection on the basis that the work does not have *potential* value. The method is interesting. I also appreciate other reviewers chiming in with their opinion.
> > >
> > > But, if the algorithm has merit, and if your paper therefore is to have the impact that it deserves too, some empirical evidence on real data is warranted. Like the authors mention, **trying this on even simple, real neuroscience data, would significantly increase the quality and potential impact of the work.** Other examples include, Neural Latents benchmark or Rotated MNIST, but there's many more the authors themselves can identify better than I can.
> > >
> > > As such, I will raise my score to a 3 (borderline reject), but I encourage the AC to think about wether the paper should undergo another round of modifications to illustrate the effectiveness of the method.

---

### Official Review · Reviewer_j9ec · 2025-07-01

**Clarity:** 2
**Significance:** 2
**Originality:** 3
**Rating:** 4
**Confidence:** 2

**Summary:**

The paper demonstrates a way to implement correlated priors over latent features in a dynamical systems model of Bayesian inference, inspired by the rodent olfactory bulb. Experimental results show that allowing nonzero correlation in the feature priors can better capture true feature distributions than assuming independence in the prior. Under the assumptions that the "sister cell" model holds, the paper shows how to estimate priors empirically from the responses of those sister cells, which could hypothetically be recorded.

**Questions:**

Can the authors provide citations to computational papers on the mammalian olfactory bulb demonstrating that its architecture as a latent-factor inference circuit generalizes across species?

In what sense does the paper's non-spiking, real-valued model of neurons incorporate interneurons vs pyramidal cells?  Interneurons would typically (admittedly, this is in cortex) refer to inhibitory cells with many-to-one connectivity, or (in the spinal cord or in very simple model organisms) simply to any neuron that is neither a sensory nor a motor neuron.

**Ethical Concerns:**

["NO or VERY MINOR ethics concerns only"]

**Final Justification:**

Upon reading the Author Response I must grant that spike-rate encoding is standard in this field, while also reducing my confidence in my final score. I hope that with the other reviewers and the area chair we can come to a more confident assessment of the paper.

**Limitations:**

The paper's model, as stated by the authors under Limitations, can only enable estimation of neural priors under the "grandmother cell" assumption, an assumption that we have fairly strong reasons not to believe in the form it is stated here: https://www.tandfonline.com/doi/full/10.1080/23273798.2016.1235279 .

**Paper Formatting Concerns:**

There are a number of section and equation labels displayed in pseudo-Latex throughout the paper's text.

**Quality:**

2

**Strengths And Weaknesses:**

Strength: mathematical fluency

Weakness: stating log-probability densities algebraically instead of as members of exponential families, parameterized in a standardized way, robs this paper of much of its clarity.  Definitions are not given in the manuscript and the primary theorem appears to be in the supplementary material.  Lines 129-137 on page 5 basically hand-waves a sizeable amount of linear algebra that should have been stated in Proposition form and then had reasonable-length proofs given in the Supplemental Material for checking.

Strength: good experimental results
Strength: despite use of the problematic "grandmother cell" assumption, decoding of priors from network activity is an interesting demonstration

---

> ### Author Rebuttal · Authors · 2025-07-30
>
> We appreciate the reviewer's feedback, which raises a number of important points, both about the biological realism and applicability of our model and its mathematical presentation. We will address the points relating to biology first.
>
> # Relation to Biology
> We would like to begin by stating that our primary aim was not to propose a new model of an olfactory bulb in a particular species, but to take inspiration from its architecture, particularly in rodents, to propose an algorithm that we believe applies more generally to other inference tasks that require correlated priors, including those beyond olfaction, encompassing natural as well as artificial systems, which we believe is of interest to the machine learning community. Hence, we avoided detailed explication of olfactory bulb anatomy and instead focused on the elements most relevant to our algorithm. However, we do believe that our algorithm provides some insight into the structure and function of the rodent olfactory bulb, hence our comparisons to experimental data from that system. Should we decide to apply our work more directly to understanding mammalian olfactory bulbs, we will be sure to take the reviewer's comments on board and highlight the connections in more biological detail.
>
> ## Q1
> The reviewer is right in their summary to point out that we speak of "the" olfactory bulb - implying that all mammalian olfactory bulbs are similar - and asks in their first question for citations of computational papers demonstrating its architecture as a latent-factor inference circuit generalises across species. We are unaware of computational papers demonstrating such cross-species generalisation. However, we point out that it is common in normative modelling of olfaction to speak of "the" mammalian olfactory bulb, rather than specialising to a particular species. For example, see the following papers, which all treat "the" olfactory bulb within the latent-factor inference framework, and none of which are from us: Koulakov, Rinberg 2011, Grabska-Barwinska et al. 2016, Kepple et al. 2019, Zavatone-Veth, Masset et al. 2023. This is because such models at present, including ours, aim primarily to explain qualitative aspects of olfactory system dynamics that depend on anatomical properties shared by many mammalian olfactory systems, such as ORNs projecting by receptor type to glomeruli, mitral/tufted cells sampling these glomeruli and projecting to the cortex, while interacting indirectly via inhibition by granule cells. As such models increase in sophistication and include additional biological details, we expect that they will correspondingly specialise their results to specific olfactory bulbs. We thank the reviewer for raising the issue of cross-species comparison, and believe that it will be fruitful to look, in future work, at how different sister cell architectures in different species may impact inference.
>
> ## Q2
> The reviewer is correct that our model incorporates non-spiking neurons, although the real-valued responses are meant to represent spike rate, in absolute terms for the 'granule cells', and relative to a baseline for the 'mitral cells'. Modeling spike rates rather than spikes is common in the field; see e.g. the first three papers cited in our response to the previous question. Our 'mitral' cells interact with inhibitory 'granule' cells. The latter are interneurons in the sense the reviewer describes: they are neither sensory nor motor neurons. They, in fact, lack axons, so cannot themselves report the results of inference beyond the bulb. This is OK when our model is viewed, as we intend, as an inference algorithm _inspired_ by the olfactory bulb, since the results of inference can be read out from the 'granule cells'. It may be problematic, however, if our model is interpreted as a description of the operation of the olfactory system, since the bulb would be unable to directly communicate the results of inference, as represented by granule cell activity, to downstream cortex - although these results could be read out indirectly through the influence of granule cells on mitral cells, which _do_ project to the cortex. Other works, e.g. that of Koulakov and Rinberg cited above, also focus on how odours can be represented by the granule cells. Such issues are perhaps why the reviewer asks whether we model interneurons or pyramidal cells. One way to resolve these issues is to think of our 'granule cells' as encapsulating a disynaptic circuit in which mitral cells project to pyramidal neurons in olfactory cortex, which (as is known anatomically) project back onto granule cells, which in turn inhibit the mitral cells. We did not delve into these important nuances because our model is an algorithm inspired by the olfactory bulb, but is not meant to describe it in detail. Should we decide to apply our model to explicating olfactory bulb structure, we will be sure to explicitly address the different roles of inhibitory interneurons in the bulb vs pyramidal cells in the cortex.
>
> # Mathematical presentation
> We now address the reviewers' points about our mathematical explication.
>
> The reviewer regretted our choice of stating log probability densities algebraically rather than as members of exponential families. We made this choice mainly to clarify the link between the log posterior being maximised (eqn 1) and the dynamics of the maximisation (eqn 2 and 3), hoping that it would illustrate how the latter equations extremize the former. We apologise for any confusion or lack of clarity that this may have caused.
>
> The reviewer is also correct that our primary Theorem, showing how the loss function extremized by the sister cell dynamics, eqn 6, can be expressed as a loss incorporating correlated priors, eqn 8. Due to space constraints and because the derivation is purely algebraic, we limited our description in the main text to mentioning that the most important step is the standard procedure of completing the square and relegated the detailed step-by-step derivation to the supplement. We again apologise for any confusion this may have caused.
>
> The reviewer also points to the "hand-waving" of a significant quantity of linear algebra in lines 129-137. Though we don't consider our description "hand-waving" as we believe the lines mentioned constitute a detailed proof, including such detail in less than 10 lines due to space constraints certainly produced an overly-compressed presentation, and we apologise for the resulting difficulty in parsing the dense text. In fact, this issue was also raised by another reviewer. In the future, we will take the reviewer's advice and provide the statement of the result as a Proposition in the main text, and relocate the proof to the supplementary material.

---

### Official Review · Reviewer_2zE9 · 2025-07-02

**Clarity:** 3
**Significance:** 3
**Originality:** 3
**Rating:** 5
**Confidence:** 4

**Summary:**

This paper develops a neural circuit model of the olfactory system that performs probabilistic inference using priors that capture correlations between latent generative factors of stimuli. To overcome the architectural challenges associated with modeling such correlations as direct connections between units representing latent variables, the authors leverage "sister cells" — a type of cell that receives copies of the receptor input — to bake this prior indirectly into their model. In this framework, the requisite coupling between latent units are captured by the strength of the synaptic connections between sister cells and latent units. The authors provide geometric intuition and a bound on the number of correlated latent causes that can be modeled. The model outperforms models that assume priors with independent latents for stimuli when inferring stimuli generated from correlated latent factors. The authors also show that the connectivity structure for a desired correlation structure exhibits degeneracy, and that different connectivity patterns provide different levels of response heterogeneity. They demonstrate that this link may be used to infer the connectivity structure from experimentally recorded responses. Finally, the authors investigate whether the responses can be used to infer the priors themselves. Under assumptions of one-hot feature representation and given a choice of inferred stimulus values, they can calculate a prior correlation matrix, which is then compared against latent factor correlations present in an ethologically relevant dataset.

**Questions:**

1. Since the objective in eq 8 uses the synaptic weights that average across sister cells (from eq 7), am I right in saying that the dynamics implied here are specified for the average sister cell activity?
2. To verify the bound derived in section 3, can you show via simulations what would happen if there are fewer sisters than input channels, and how the number of features grows with the number of sister cells?
3. If the stimuli were composed of features where only a portion of the features are correlated (with the rest being independent), how would inference with correlated priors look like? Similarly, it would be useful to show the converse of section 4, that if the stimuli were composed of fully independent features, performing inference using correlated priors would still recover the independent factors (i.e. match the performance of a standard model with independent priors).
4. In section 6, it is surprising that there is a mismatch between correlated prior estimates and the empirical observation of correlations in monomolecular components. Is this purely a function of assuming grandmother cells, or are there other assumptions that lead to this mismatch? Conversely, under which regimes would I recover the observations even under a grandmother cell assumption?

**Ethical Concerns:**

["NO or VERY MINOR ethics concerns only"]

**Final Justification:**

The authors have provided thorough responses, both to the questions that I raised and to the identified weaknesses. I have also read the other reviews and in my opinion further benchmarking is not necessary (aGi5). The authors have already demonstrated that their method can infer correlated features in synthetic data (in Fig 2) and compared their estimate of the correlated priors to a naturalistic database of odor latents (in Fig 4). While the correlations in the model's priors do not match those observed in data, in the rebuttal the authors have expounded on the reasons as to why this might be the case. Overall, I am satisfied with the responses provided by the authors, and maintain my decision to accept the paper, with the same level of confidence as before.

**Limitations:**

yes

**Paper Formatting Concerns:**

I'm not entirely sure if the latex labels in the margins is a formatting concern, but it is unusual.

**Quality:**

4

**Strengths And Weaknesses:**

Strengths

[+] Quality: The model successfully overcomes the problem initially proposed at the onset of the paper, as to whether correlations can be included in the prior without invoking direct interactions between latent cause units, which is both empirically unsupported and computationally impractical. The mathematical derivations are sound and well reasoned. I also appreciated the attempts to form a tighter connection to empirical data, both by comparing model predictions with experimentally measured unit responses, and by comparing the model correlation matrix against the empirical correlations extracted from the ethologically relevant odor dataset.

[+] Clarity: It was clear which parts were assumptions, which were novel contributions, and which were consequences of the data/model. The investigatory questions following the model formulation were thoughtful and flowed logically, including a sanity check to see if this model outperforms the vanilla model with a prior of independent latent factors.

[+] Significance: This model proposes a novel way to incorporate correlations in the latent factors in the prior of a neural circuit model of the olfactory system, and provides what are the necessary conditions for using sister cells to encode n correlated features in the prior.

[+] Originality: while encoding priors and performing inference via fixed points of a dynamical system is not new, this work provides a novel way to induce implicit correlated priors in a circuit via the connectivity structure between latent units and neurons that receive shared receptor input.

Weaknesses

[-] It would be useful to have a little bit more context to connect Eq 1 to Eq 2 and 3, otherwise ref 6 must be read to understand the model setup.

[-] Some parts of section 3 (finding the bound on the number of features, and finding the set of all possible solutions to W and B) were overly dense, especially since not all of the conclusions were necessary for the rest of the paper and/or verified via simulations. I believe it could be tightened by keeping only the essentials and punchlines of the derivations and moving the rest to the appendix.

[-] Clarity: Some of the wording explaining why it is reasonable to first zero out the affinities is unclear. There are also a couple of typos; line 218: "this the" → "this is the"; line 223: "compute to" → "compute the"; line 266: "quantitavely" → "quantitatively"; line 273: "esimated" → "estimated"; line 303: "statistcs" → "statistics".

---

> ### Author Rebuttal · Authors · 2025-07-30
>
> We thank the reviewer for their detailed and thoughtful comments on our submission, and apologise for any lack of clarity in our presentation. We will first address the weaknesses the reviewer raised, and then their questions.
>
> # Weaknesses
> - We regret that the connection between the log posterior in Eq 1, and the dynamics in Eq 2 and 3 wasn't clearer. Although ref 6 does indeed provide more context, the fundamental connection is simply gradient ascent: the 'granule cell' dynamics in Eq 3 ascend the gradient of the log posterior in Eq 1 relative to the latents, employing the 'mitral cells' in Eq 2 as auxiliary variables to compute a component of the gradient.
>
> - Density of parts of section 3: We apologise for the compressed presentation in this section - it was an issue raised by another reviewer as well. Next time we will take the reviewer's suggestion and leave only the essentials in the main text, moving the rest, decompressed, to the appendix.
>
> - Zeroing out affinities: We apologise that the reasoning behind this subtle point was not completely clear. In that section, we are determining how to set connectivities to achieve a particular covariance of the weights. These covariances are computed over sisters, and measure covariability relative to the mean value of the weights computed over sisters. Thus, changing this mean value, which equals the affinities, does not affect the covariances. Therefore, to simplify the derivation, we can first pretend the affinities are zero and compute the weights that achieve the desired covariance. We can then add the correct affinities to all the weights, without changing the covariances achieved by the weights.
>
> - Spelling mistakes: We apologise for these and thank the reviewer for pointing them out.
>
> # Questions
> We first would like to say that we are grateful for the reviewer's thoughtful questions, which reveal their engagement with our submission.
>
> ## Q1
> The objective in eq 8 indeed uses synaptic weights that are averaged across sister cells. And indeed, if one averages the sister cell dynamics in eq 4, one see that the dynamics of the resulting average sister cell activity use the weights averaged across sister cells. However, a key aspect of those dynamics is how they are influenced by the granule cells (x_j). If we replace the sister cells with their average, then the information about correlations in the prior can no longer be represented in the different activity levels among sisters, since these have been replaced by an average. Instead, it must be supplied by direct interactions between the granule cells, similar to eqn 3. Therefore, the loss in eq. 8 can correspond to dynamics using average sister cells, using the average weights, interacting with granule cells that interact directly in order to implement correlations in the prior.
>
> ## Q2
> Verifying the bound would be an important addition to the manuscript. The number of sister cells S is always at least equal to the number of input channels M, since there is at least one mitral cell per input channel (otherwise the information from that channel would be unavailable to the system), and we call all mitral cells sampling a channel sisters, even, perhaps confusingly, when there is only one such cell. One way to verify the bound would be to directly optimise the weights to achieve a desired set of affinities and covariances. We currently construct such weights using the geometric approach that we used to derive the bound. However, the optimisation approach could be made agnostic to our derivation, and to merely search for sister cell weights that have the relationships we require. If our bound is correct, then we should find that when the dependencies we require are among more latents than our sister cell budget allows, the optimisation fails because the problem is not feasible. Therefore, plotting the frequency of such failures as we increase the number of dependencies encoded could serve as a partial verification of our bound.
>
> ##  Q3
> The configuration of correlations the reviewer wonders about is, in fact, already the case for the simulation results in Figure 2, for which only 5 of the features (the first 5) are correlated in the prior, and the rest are independent.
>
> Showing that the model using correlated priors can still perform inference when the latents are, in fact, independent, is a good suggestion, and we could certainly include such results. However, we would expect that performance in that setting would be inferior to that using a prior that assumed independence, since the correlations in our prior would bias inference to reflect such correlations, when in fact none would exist. This would essentially mirror the results shown in Figure 2C, and would reflect the reduction in inferential accuracy when the assumed prior does not match the statistics of the data.
>
> ## Q4
> There could be a number of reasons for the mismatches we observed in Figure 4, relating both to our model and to the available data on the frequencies with which pairs of molecules are present in natural odours. First, the predictions in those panels require assuming that our model of sister cell dynamics in equations 4 and 5 are correct, and that the grandmother cell assumption is correct. Both of these are at best approximations.
>
> Even under the grandmother cell assumption, the predictions require knowing the inferred concentration of each odour (c_j and c_k in Eq 10). A given data point on the observed correlation of two monomolecular odours would provide a value for Cjk in Eq 10, and thereby constrain the product of concentrations on the right-hand side of that equation. We only had values for 6 pairs of odours, so rather than fitting the concentration values to optimise the fit, we opted to make three different assumptions about the inferred concentrations and show the results in the panels of Figure 4. If we had at least as many pairwise datapoints as we have monomolecular odours, then we would have enough data to fit all the concentrations, up to an overall scaling factor, and would then hope to improve our match to the observations.
>
> Finally, assuming our model is correct and we had sufficient data on odour co-occurrence statistics, we would only expect a match if those statistics are those of the environment in which the animals whose bulbar response data lived. Thus, the ideal experiment would be to raise animals in an environment in which the co-occurrence statistics are known, and then to use our approach to estimate these co-occurence statistics from the sister cell responses.

---

> > ### Comment · Reviewer_2zE9 · 2025-08-07
> >
> > I thank the authors for their thorough responses, both to the questions that I raised and to the identified weaknesses. In response to their answer to question 2: while I would personally be interested in seeing the numerical validation of the bound using the method proposed by the authors, I do not believe it is necessary at this stage. Overall I am satisfied with the responses provided by the authors and maintain my decision to **accept** the paper.
> >
> > I have also read the other reviews and in my opinion further benchmarking is not necessary (aGi5). The authors have already demonstrated that their method can infer correlated features in synthetic data (in Fig 2) and compared their estimate of the correlated priors to a naturalistic database of odor latents (in Fig 4). While the correlations in the model's priors do not match those observed in data, in the rebuttal the authors have expounded on the reasons as to why this might be the case.

---

### Official Review · Reviewer_W7wp · 2025-07-04

**Clarity:** 2
**Significance:** 2
**Originality:** 2
**Rating:** 5
**Confidence:** 1

**Summary:**

This paper presents a mathematical formulation inspired by sister cells, demonstrating how the connectivity between these and the latent units can enhance inference performance when latent variables are correlated.

I have minimal knowledge on this topic and am not up to date with the current state of this research. So, please read my criticism below with a grain of salt.

**Questions:**

Q1)  Can authors show how much boost they get by optimizing the sparse connectivities?
Q2) Can authors sketch the inference steps if the receptor responses were not modelled as linear/isotropic Gaussian observations? What are the challenges and what solutions are there to overcome those?

**Ethical Concerns:**

["NO or VERY MINOR ethics concerns only"]

**Final Justification:**

The authors' responses addressed my concerns during the rebuttal period. I keep my original score, as it reflects the state of the paper post-rebuttal as well.

**Limitations:**

Yes.

**Paper Formatting Concerns:**

I don't know if it's allowed to add those "naming" on each equations and paragraphs with the italic font on the right side of each paragraph. I guess it's useful to go back and find those things easily when necessary, but I am unsure if this is against the formatting rule.

**Quality:**

4

**Strengths And Weaknesses:**

Strengths:
1. Extensive experimental study
2. Solid mathematical modelling
3. Well-explained, very clearly written paper

Weaknesses:
1. As mentioned by the authors, the receptor responses were modelled as linear/isotropic Gaussian observations, which can be viewed too simplistic.
2. Sparse connectivity can also be optimized by introducing hyperparameters (as mentioned in the limitation section).

---

> ### Author Rebuttal · Authors · 2025-07-30
>
> We thank the reviewer for their comments. Below we provide answers to the two questions they raised.
>
> Q1) The reviewer raises an important point about the role of sparsity in our model. As we show in the main text, many connectivities encode the same receptor affinities and molecular coactivation statistics, spanning the spectrum from sparse to dense. Therefore, the computational boost when using sparse connectivity is not in the final result of inference, as that would be the same for all such connectivities. The advantages gained from sparsity would principally be in the sparsity itself: fewer synapses would need to be adjusted to encode the same statistics. There may be dynamical advantages wherein sparser connectivities arrive at the solution faster than denser ones, but we have not explored this important question. Our main motivation for exploring the role of sparsity was the empirical observation in the olfactory system that sister cells can have similar responses to some odours, and different responses to others. We achieved a comparable level of heterogeneity using a weighted sparsity solution, where sisters from some glomeruli had denser connectivity to the granule cells than others, and therefore suggest that the olfactory system may operate in this regime. This sparsity may be a result of learning - work by others has shown that learning in neural systems using 'exponentiated gradients' produces synapses that obey Dale's law, so don't change sign during the learning process, is competitive or superior to standard gradient descent on many tasks, and produces sparse, log-normally distributed synaptic weights. Such learning in the olfactory system could produce the sparse connectivity that appears to be necessary to reproduce the response heterogeneity observed empirically. We plan to address this point in future work on how connectivity encoding odour statistics, which we constructed by hand in this manuscript, can be learned.
>
> Q2) The reviewer raises an important point that we also mention in our "Limitations" section. The way we encode second-order odour statistics relies on the linear Gaussian aspect of our model, which allows 'completing the square' and an interpretation of the sister cell loss as encoding receptor affinities and molecular coactivation statistics. We have not yet explored whether deviations from such a simple model would readily yield a similar encoding of odour statistics. We speculate that non-isotropy would be easy to incorporate, as differences in variance among the glomeruli could be analytically equalised by absorption into synaptic weights and the strengths of glomerular inputs, perhaps at the cost of some degrees of freedom. Deviations from the linear-Gaussian model, however, seem much more difficult to accommodate. For example, a more accurate model of receptor responses would use Poisson statistics for the spike counts. However, at least in our back-of-the-envelope derivations, which used a linear-Poisson model, we were unable to readily encode higher-order statistics in the connectivity, though it's possible that with further work, we could do so. Therefore, at this point, our results should be interpreted as limited to linear Gaussian models.

---

> > ### Comment · Reviewer_W7wp · 2025-08-04
> >
> > I thank the authors for their responses, which addressed my concerns. I keep my original score, as in my opinion, it reflects the state of the paper post-rebuttal as well.

---

### Note · Authors · 2025-08-13

We are grateful to all reviewers for their thoughtful and constructive engagement with our submission. We are encouraged that multiple reviewers appreciated our clear mathematical motivation, the novelty of our biologically inspired approach, and our attempts to meaningfully link our results back to experimental data.

Our desire to reveal as much of the mathematics behind our approach as possible in the Main Text resulted in some reviewers finding parts of the mathematical exposition quite dense. We apologise for this, and in the future will follow the reviewers' advice to relegate most such details to the Supplementary Information.

We have also carefully considered the request from Reviewer aGi5 for further benchmarking. We certainly agree that extensive benchmarking is necessary to determine the full capabilities of a proposed algorithm, particularly for end-to-end learning systems composed of many complex components. However, as we explained in our rebuttal, our current work aims primarily to introduce and analyse a new, biologically inspired, computational mechanism. We therefore focused on mathematical derivations and targeted tests demonstrating its properties, such as recovering correlated features in synthetic data (Fig. 2), qualitative explication of the diversity of experimentally recorded sister cell responses (Fig. 3), and how it can be used to infer the correlated priors animals use from their neural responses (Fig. 4), including a comparison of the latter to the (limited) relevant publicly available data. We thus echo the sentiments of Reviewers j9Ec and 2zE9 and hope that further benchmarking, while an important avenue for future work, is not necessary to validate our present contribution.

We once again thank the reviewers for their time and for the valuable feedback they have provided.

---

### Decision · Program_Chairs · 2025-09-17

**Decision:**

Accept (poster)

**Comment:**

This paper introduces a novel circuit model for performing probabilistic inference with correlated priors using a framework called "sister cells". It shows both theoretically and empirically that this architecture outperforms models with independent priors, and links connectivity structure to observed neural response heterogeneity. Three of the four reviewers felt this work was above threshold for acceptance, and I ultimately concur with their assessment that the paper makes a worthwhile contribution to the literature and should be accepted.  Congratulations! Please be sure to address all reviewer comments and criticisms in the final manuscript.